# Molecular Drivers of Myelodysplastic Neoplasms (MDS)—Classification and Prognostic Relevance

**DOI:** 10.3390/cells12040627

**Published:** 2023-02-15

**Authors:** Fieke W. Hoff, Yazan F. Madanat

**Affiliations:** 1Department of Internal Medicine, UT Southwestern Medical Center, Dallas, TX 75390-8565, USA; 2Harold C. Simmons Comprehensive Cancer Center, UT Southwestern Medical Center, Dallas, TX 75390-8565, USA

**Keywords:** myelodysplastic neoplasms, molecular drivers, genetics, prognostication, classification

## Abstract

Myelodysplastic neoplasms (MDS) form a broad spectrum of clonal myeloid malignancies arising from hematopoietic stem cells that are characterized by progressive and refractory cytopenia and morphological dysplasia. Recent advances in unraveling the underlying pathogenesis of MDS have led to the identification of molecular drivers and secondary genetic events. With the overall goal of classifying patients into relevant disease entities that can aid to predict clinical outcomes and make therapeutic decisions, several MDS classification models (e.g., French–American–British, World Health Organization, and International Consensus Classification) as well as prognostication models (e.g., International Prognostic Scoring system (IPSS), the revised IPSS (IPSS-R), and the molecular IPSS (IPSS-M)), have been developed. The IPSS-M is the first model that incorporates molecular data for individual genes and facilitates better prediction of clinical outcome parameters compared to older versions of this model (i.e., overall survival, disease progression, and leukemia-free survival). Comprehensive classification and accurate risk prediction largely depend on the integration of genetic mutations that drive the disease, which is crucial to improve the diagnostic work-up, guide treatment decision making, and direct novel therapeutic options. In this review, we summarize the most common cytogenetic and genomic drivers of MDS and how they impact MDS prognosis and treatment decisions.

## 1. Introduction

Myelodysplastic neoplasms (MDS) are a spectrum of clonal myeloid stem cells malignancies, which are characterized by ineffective hematopoiesis resulting in various degrees of progressive and refractory cytopenia and morphological dysplasia in one or more cell lines (i.e., erythroid, granulocytic, and/or megakaryocytic dysplasia) [1]. MDS is a disease of the older population with a median age of diagnosis of over 70 years. Approximately 30% of MDS patients will progress to acute myeloid leukemia (AML) [2]. The molecular landscape of MDS has not yet been fully elucidated, but recent advances have led to an improvement in the understanding of molecular pathogenesis. Studies have demonstrated that a large portion of the genetic mutations present in MDS have also been identified in healthy individuals, and while some individuals do have evidence of mild cytopenia, they otherwise do not fulfill the criteria of MDS. When present at a variant allele frequency (VAF) ≥2% and in the absence of blood count abnormalities, these mutations are incorporated under the term “clonal hematopoiesis of indeterminate potential” (CHIP). In contrast, when detected in the setting of cytopenia but without morphologic evidence of dysplasia, they fulfill the criteria for the diagnosis of clonal cytopenia of undetermined significance (CCUS) [1,3]. MDS frequently develops in the setting of CHIP or CCUS with the accumulation of somatic mutations in the hematopoietic stem cells over time.

MDS is defined as having persistent and unexplained cytopenia of at least one lineage of hematopoietic cells in combination with dysplastic features in ≥10% of the nucleated myeloid cells diagnosed on a bone marrow biopsy and/or recurrent genetic abnormalities that provide presumptive evidence of MDS [2,4]. Further characterization of the molecular landscape of MDS is crucial to improve understanding of the pathogenesis of MDS and for classification into clinically relevant MDS entities to enhance outcome prognostication and to aid in medical decision making. The aim of this review is to discuss recent advances in disease classification and novel MDS risk-scoring systems, and to summarize the most common genomic abnormalities in MDS and their impact on disease pathogenesis and prognostication.

## 2. Classification of MDS

### 2.1. French–American–British (FAB) Classification

MDS encompasses a heterogeneous group of diseases with a wide range of clinical characteristics and cytologic and molecular features. Several refined classification and scoring systems have been developed to define more homogeneous subgroups of MDS. The FAB classification was the first classification system for MDS and was published in 1982 by a group of pathologists from France, the United States of America, and Britain [5]. It defined MDS as refractory anemia and categorized it into five different groups based on morphologic criteria and percentage of myeloid blasts: refractory anemia (RA), RA with ringed sideroblasts (RARS), RA with an excess of blasts (RAEB) (defined as <5% peripheral blood (PB) or 5–19% bone marrow (BM) myeloblasts), chronic myelomonocytic leukemia, and RAEB in transformation (RAEB-T, defined as ≥5% PB or 20–29% BM myeloblasts or presence of Auer rods). The FAB classification has been used as the gold standard for almost 20 years, but with the identification of additional prognostic factors, the MDS classification was updated by the World Health Organization (WHO) [6].

### 2.2. WHO Classification

The WHO classification was introduced in 2001 (3rd edition) and it was the first classification system that incorporated genetic information. It takes into account the number of dysplastic lineages, the presence or absence of ring sideroblasts, the percentage of BM and PB blasts, and cytogenetic abnormalities. Revised versions of the WHO classification were published in 2008 and 2016 and incorporated more knowledge on clinical, morphological, immunophenotypic, and genetic features to further refine the classification of clinically relevant disease entities [2]. In 2022, clonal hematopoiesis was added as a category of precursor myeloid disease state and was formally recognized as CHIP or CCUS. CCUS is defined as CHIP in the presence of one or more persistent cytopenias that are otherwise unexplained [7,8]. When compared to older versions, the threshold of dysplasia remained at 10% in any hematopoietic lineage, but MDS is now grouped into two entities—those having defined genetic abnormalities and those that are morphologically defined. The incorporation of CCUS obviates the need for “unclassifiable MDS (MDS-U)” or “NOS”. The WHO 2022 classification and defining features of MDS include:
MDS with defining genetic abnormalities;
○MDS with low blasts and isolated 5q deletion (MDS-5q);○MDS with low blasts and *SF3B1* mutation (MDS-*SF3B1*);○MDS with biallelic *TP53* inactivation (MDS-bi*TP53*);
MDS, morphologically defined;
○MDS with low blasts (MDS-LB);○MDS, hypoplastic (MDS-h);○MDS with increased blasts (MDS-IB);
▪MDS-IB1;▪MDS-IB2;▪MDS with fibrosis (MDS-f).

### 2.3. International Consensus Classification (ICC)

The ICC is a clinical advisory committee that includes many authors of the prior WHO editions but is no longer affiliated with the WHO. In 2022, the ICC came up with a new classification by introducing new disease entities and redefining criteria for existing MDS classes compared to the WHO classification [9]. Key differences include the reclassification of MDS with excess blasts (EBs) (5–9% in the BM and/or 2–9% in the PB) for which there is now only one MDS-EB subtype and MDS with 10–19% blasts in the PB or BM without AML defining genetic lesions that is changed to MDS/AML. Patients with MDS/AML may become eligible for both MDS and AML trials, which will further optimize treatment options. Moreover, genetic categories of MDS with mutated *SF3B1* (VAF ≥10%) and MDS with multihit *TP53* (defined as two distinct *TP53* mutations (each VAF ≥ 10%) or as a single *TP53* mutation with either a 17p deletion on cytogenetics, a VAF of ≥ 50%, or a copy-neutral LOH at the 17p *TP53* locus) have been introduced. The ICC classification of MDS includes:MDS with mutated *SF3B1* (MDS-*SF3B1*);MDS with del(5q) [MDS-del5q)];MDS, NOS without dysplasia;MDS, NOS with single lineage dysplasia;MDS, NOS with multilineage dysplasia;MDS with excess blasts (MDS-EB);MDS/AML.

An overview of similarities and differences between the WHO 2022 and ICC 2022 classifications is shown in Table 1.

### 2.4. Internal Prognostic Scoring System (IPSS)

Given the imprecision of classification with regard to clinical outcomes, the International Prognostic Scoring System (IPSS) MDS prognostication system was created for primary untreated MDS (Table 2) [10,11,12]. Two refinements of this model have been published since its introduction in 1997; the IPSS-revised (IPSS-R) in 2012 and, most recently, the IPSS-Molecular (IPSS-M) in 2022. The original IPSS considered relatively limited clinical and cytogenetic information (percentage of BM blasts, karyotype, and number of cytopenias) and classified patients into low (IPSS score 0), intermediate-1 (score 0.5–1), intermediate-2 (score 1.5–2), and high-risk MDS (score 2.5–3.5). Given the ease of calculating the IPSS score, many clinical trials to date continue to use this score as part of the inclusion criteria to risk stratify patients into lower-risk MDS (low and intermediate-1 risk) and higher-risk MDS (intermediate-2 and high). The newer IPSS-R included further risk assessment based on cytogenetics, refined cut-offs of the percentage of myeloblasts, and the degree of cytopenia, resulting in five risk categories: very low (IPSS-R score ≤ 1.5), low (score >1.5–3), intermediate (score 3–4.5), high (>4.5–6) and very high (>6). It is debatable whether patients with intermediate-risk disease should be considered low or high risk, and the decision of how to manage these patients is therefore often difficult, individualized, and context-dependent. A single-center comparative analysis (*n* = 128) published by Warlick et al. (ASH 2012, Abstract #3841) showed that IPSS-R intermediate-risk patients would be reclassified as low/intermediate-1 (64%) or intermediate-2 risk (36%) in the IPSS. A larger retrospective study evaluated IPSS-R intermediate-risk MDS patients (*n* = 298) and subclassified patients based on clinical factors that influence survival into favorable-intermediate and intermediate-adverse risk [13]. They showed a significant difference in median survival between both cohorts, emphasizing variable outcomes in patients with IPSS-R intermediate-risk disease. Given that treatments for “lower risk” and “higher risk” are substantially different, varying from observation to hematopoietic stem cell transplantation, we should, therefore, carefully consider treatment approaches for intermediate-risk patients that are not simply based on an IPSS-R risk score cut-off of 3.5 as currently used in MDS guidelines [14].

While the IPSS-R relies on hematologic and cytogenetic features, it does not include genomic data. Based on data showing that genetic mutations harbor prognostic information [15,16,17], the IPSS-M, for the first time, includes molecular data for 31 genes (Figure 1), which resulted in six prognostic risk groups [10,18]. The median OS ranged from 10.6 years in the very-low-risk group to a median OS of 1.0 year in the very-high-risk group. IPSS-M performed better at predicting leukemia-free survival (LFS), AML transformation, and overall survival (OS) compared to the IPSS-R. The IPSS-M re-stratified nearly half of the patients with MDS, of which 74% were upstaged and 26% were down-staged to a lower-risk group and which is also applicable to secondary/therapy-related MDS. The number of prognostic genes is significantly higher than previously reported in other studies, likely due to the high number of patient samples included in the analysis to build the IPSS-M model (*n* = 2957). Even underrepresented genes in MDS, including NPM1, FLT3, and MLL-PTD, maintained independent prognostic value, as they do behave more like AML. In the ICC 2022 and WHO 2022, AML can be diagnosed with ≥10% blasts in the ICC and regardless of blast count in the WHO.

The downside of the IPSS-M classification, however, is that while genomic testing for MDS is considered the standard of care in the United States, it is not readily available in certain parts of the world, limiting the worldwide applicability of IPSS-M. Moreover, given the complexity of this scoring system, an online calculator is necessary to determine the risk score. The clinical IPSS-M web calculator can be found here: http://mds-risk-model.com/ (accessed on 10 January 2023).

## 3. Cytogenetic and Molecular Landscape of MDS

MDS is a clonal disorder that starts with the initiation of somatic mutations that occur in the genome of the multipotent hematopoietic stem cell (HSC). Mutations that provide growth and survival benefit at the level of the HSC and enhance self-renewal lead to the accumulation of clonal hematopoiesis over time, resulting in abnormal progenitor and precursor cells. Given the selective survival advantage of these initiation events over wild-type cells, these somatic mutations are termed “driver mutations”. Advanced high-throughput sequencing technologies have led to the discovery of recurrent chromosomal abnormalities, mutations that alter the expression of individual genes, and epigenetic abnormalities [16,18,19].

### 3.1. Recurrent Cytogenetic Abnormalities in MDS

Approximately half of the patients with MDS harbor recurrent chromosomal abnormalities affecting copy number alteration (e.g., deletion, monosomy, or trisomy) or, more rarely, leading to a structural change (e.g., balanced translocation or inversion). The most common MDS-defining chromosomal abnormalities are deletion 5q (10–15%), monosomy 7/deletion 7q (10%), trisomy 8 (10%), and deletion 20q (5%) [20]. Up to 30% of MDS patients exhibit a complex karyotype (≥3 cytogenetic abnormalities), which is associated with a higher risk of progression to AML and a very poor prognosis. Complex karyotypes with more than three abnormalities has been found to be distinct from those with three abnormalities and is associated with even inferior outcome with a median OS of 1.5 vs. 0.7 years [11]. The aforementioned abnormalities, in addition to deletion 12p/addition (12p), isochromosome (17q), monosomy 17/addition or deletion of 17p, and/or idic(X)(q13), define MDS per the international consensus classification, and when detected in AML, would make a diagnosis of AML with myelodysplasia-related cytogenetic abnormalities [9].

#### 3.1.1. Deletion 5q

Deletion 5q was first described almost 50 years ago and is the most common cytogenetic abnormality that is present in 10–15% of MDS patients, with a higher incidence in therapy-related or secondary MDS [21,22]. If isolated or in the setting of one additional cytogenetic abnormality apart from monosomy 7/del(7q), deletion 5q is associated with a favorable prognosis with a 15% probability to develop transformation in AML after 5 years. Patients often present with various degrees of refractory cytopenia and blast count <5%, and isolated 5q typically presents in older women with a median age of >70 years. Isolated del(5q) has a reported 6-year OS rate of 67% and PFS of 53%, and it has been shown to have high response rates to treatment with lenalidomide for which it has regulatory FDA approval [21,23]. However, when deletion 5q is present in the context of excessive blasts or other cytogenetic abnormalities, it has a sixfold higher rate of progression to AML [24]. Multiple gene loci present on the long arm of chromosome 5 contribute to the clinical picture. For instance, haploinsufficiency of ribosomal protein S14 (*RPS14*) contributes to the development of dyserythropoeisis via activation of the p53 pathway [25]. Haploinsufficiency of casein kinase 1 alpha 1 (*CSNK1A1*) plays a role in the initiation of clonal expansion via deregulation of the WNT/beta-catenin pathway [26].

#### 3.1.2. Monosomy 7 and Deletion 7q

The second most common cytogenetic abnormality in MDS is monosomy 7 or deletion 7q, which occurs in 10% of patients with de novo MDS and in about 50% of patients with treatment-related MDS [27]. Monosomy 7 is more prevalent than deletion 7q and is associated with a worse prognosis. Both monosomy 7 and deletion 7q are caused by completely different mechanisms of chromosome dis-segregation in mitosis versus chromosome rearrangement, respectively. High-throughput sequencing technologies have identified gene mutations associated with chromosome 7 anomalies, including *SAMD9*, *SAMD9L*, *EZH2*, and *MLL3* [27,28,29,30].

#### 3.1.3. Trisomy 8

Trisomy 8 is found in approximately 10% of MDS patients and is considered an intermediate risk with a median OS of 6 years [31]. It is thought to be a secondary or late event in the MDS transformation. While the precise mechanism in the tumorigenesis process remains unclear, gain of chromosome 8 has been shown to confer more resistance to apoptosis by upregulation of antiapoptotic genes present on chromosome 8 as well as overexpression of the *MYC* oncogene [32].

#### 3.1.4. Deletion 20q

Deletion 20q occurs in 5% of MDS patients and it often appears as a major clone at diagnosis. While it is considered to have a favorable risk prognosis, the development of deletion 20q as a minor clone during stages of the disease can precede disease progression and is therefore associated with poor prognosis [33]. *ASXL1* mutation co-occurs in 30% of the patients with deletion 20q and negatively impacts prognosis [34].

#### 3.1.5. Other Cytogenetic Abnormalities

Other less commonly present chromosomal abnormalities include -Y, del(3q), del(9q), 13/del(13q), del(11q)/t(11q), del(12p)/t(12p), 17/del(17p)/i(17q), +19/t(19), and idic(Xq13). The pathogenesis of more rare cytogenetic abnormalities is largely unknown, but the advancements in genetic technology may provide deeper insights into the pathophysiology and predict the prognosis of patients with MDS [35]. As an example, while monosomal karyotype (MK), defined by the presence of at least two separate autosomal monosomies or one monosomy plus one or more structural abnormalities, has been associated with adverse prognosis in AML, Schanz et al. revealed that MK was not independently associated with prognosis in MDS and that the distinction between MK+ and MK- did not add any prognostic information [36]. Their study had over 400 patients, half of whom were identified as having MK. Moreover, while the number of abnormalities (≤4 vs. >4 abnormalities) was associated with OS, occurring more frequently in MK+ compared to MK− MDS, the impact on prognosis was independent of the presence of MK. Therefore, complex karyotype rather than monosomal karyotype is incorporated in the IPSS-R risk model as a clinical variable.

### 3.2. Recurrent Gene Mutations in MDS

While around half of the MDS patients harbor one or multiple chromosomal changes, more than 90% of the patients have mutations that alter the sequence and function of at least one oncogene or tumor suppressor gene. Using techniques including next-generation sequencing, only a few gene mutations were identified to be present in >10% of the patients, and most patients harbor 2–4 different gene mutations [16]. MDS patients also commonly have abnormal epigenetic profiles, leading to changes in their gene expression. Based on sequencing studies that calculate the VAF of genes, mutations in splicing factors and epigenetic modifiers were found to occur early in the evolution of MDS and mutations in transcription factors were found to occur either as early or late events. Hence, *TET2*, *DNMT3A*, *SF3B1*, *ASXL1*, *TP53*, and *JAK2* are the most mutated genes underlying CHIP and CCUS [16]. The commonly mutated genes and their implicated biological pathways are shown in Table 3.

#### 3.2.1. RNA Splicing

Spliceosomes are formed by five small nuclear ribonucleoproteins (snRNP) and their associated proteins and are important in the process of removing noncoding regions of the mRNA before translation [37]. Alternative splicing occurs in >90% of the protein-coding genes, allowing the production of multiple mRNA transcripts and distinct protein isoforms. Mutations in genes encoding for spliceosomal proteins in MDS lead to aberrant 3′ splice side recognition contributing to tumorigenesis. The most frequently mutated genes are *SF3B1*, *SRSF2*, *U2AF1*, and *ZRZR2*. They are present in about 60% of MDS patients and they arise mutually exclusive of each other [38,39]. While mutational targets overall are largely similar between MDS and primary AML, mutations in the spliceosomes are overrepresented in MDS.

*SF3B1* is the most frequently mutated spliceosome mutation in MDS, found in up to 30% of MDS patients [40,41]. The *SF3B1* gene encodes for splicing factor 3b, which is a member of the U2 snRNP complex and is thought to be an initiating genetic event in MDS [41,42]. It is present in >80% of MDS patients with ring sideroblasts (MDS-RS) but rare in other MDS subtypes. *SF3B1* is associated with favorable clinical outcomes with a low propensity to progress to AML. Based on the clinical implications in terms of risk stratification and therapeutic decision making, the 2016 WHO classification has included *SF3B1*-mutant MDS as a diagnostic criterion for MDS-RS with a positive predictive value of 98% [41]. Only 5% of the *SF3B1* mutated patients have poor or very poor cytogenetic risk groups [40]. The median OS of patients with mutated *SF3B1* is 79 months vs. 53 months in wild-type *SF3B1*, with progression to AML occurring in 7% of the patients after a median follow-up of more than nine years. In patients with the *SF3B1* mutation and MDS-RS with either single or multilineage dysplasia, the median OS is 106 and 82 months, respectively. Notably, *SF3B1* in the presence of more than one additional aberration (particularly *RUNX1* mutation) or in combination with del(5q) is associated with a dismal prognosis [40]. Currently, luspatercept is approved for MDS-RS for transfusion-dependent anemia post erythropoietin-stimulating agents (ESAs) or for patients who are unlikely to benefit from ESA therapy [43].

The *SRSF2* mutation is present in approximately 15% of MDS patients and, in contrast to the *SF3B1* mutation, it is associated with a worse prognosis and high transformation rate to AML [44]. *SRSF2* encodes for serine/arginine-rich splicing factor 2 protein, and mutated *SRSF2* causes alteration of the mRNA recognition, resulting in mis-splicing of key transcriptional regulators [45]. The *SRSF2* mutation often presents with dysplastic features of granulopoiesis and megakaryopoiesis. Gene mutation of *U2AF1* occurs in 10–15% and results in RNA splicing dysfunction [46]. Studies agree that *U2AF1* may predict poor prognosis with a higher risk of leukemic transformation, and *U2AF1* is one of the prognostic gene mutations included in the IPSS-M (Figure 1) [10]. *ZRSR2* is less frequently present in MDS (5–10%) and alters the splice site recognition of the pre-mRNA. The impact of *ZRSR2* on clinical outcome remains unknown [44].

#### 3.2.2. DNA Methylation

DNA methylation (CpG methylation) exerts a key role in normal differentiation and proliferation of the HSC. CpG islands are regions with a high frequency of CpG sites and are regulatory units present in promoter regions of 60–70% of the genes. Changes in DNA methylation contribute to altered gene expression without sequence mutations of the genomic DNA. Several genes that play a role in DNA methylation (e.g., *TET2*, *DNMT3A*, *IDH1*, and *IDH2*) are frequently mutated in MDS and cause global as well as gene-specific hypermethylation, resulting in silencing of tumor suppressor genes or genes involved in DNA repair. Like mutations involved in RNA splicing, mutations that affect DNA methylation typically occur early in the development of MDS.

As an example, de novo DNA methyltransferase (*DNMT3A*) is a member of the *DNMT* family that adds a methyl group to cytosine in CpG dinucleotides. *DNMT3A* is mutated in 15% of MDS patients and is frequently present in CHIP. It is associated with an increased risk of leukemic evolution and inferior prognosis [10]. Most DNMT3A mutations reside in the catalytic domain of the methyltransferase domain of *DNMT3A*, especially at the amino acid R882 locus, which results in reduced methyltransferase activity of the protein due to defective DNA binding and impaired CpG recognition [47]. It is mutated in approximately 20% of AML patients, in 5–10% of MDS cases, and in 60% of patients with CHIP [48,49]. Interestingly, while *DNMT3A* R882 mutations were found to be enriched in AML (~50% of all *DNMT3A* mutations), they are decreased in frequency in CHIP (~10%) and other myeloid neoplasms including MDS (~25–30%). Moreover, as observed with AML, MDS patients with R882 mutations are found to have a significantly worse overall prognosis and a more rapid progression to leukemia than patients with non-R882 *DNMT3A* mutations.

In contrast, *TET2* leads to hypermethylation via a different mechanism and is mutated in 20–30% of MDS patients. It is responsible for alpha-ketoglutarate (α-KG)-dependent catalyzation of hydroxylation of 5-methyl-cytosine to hydroxymethyl-cytosine, promoting DNA methylation [50]. The prognostic implication of *TET2* mutation remains unclear. *IDH1* and *IDH2* are enzymes that catalyze the oxidative decarboxylation of isocitrate to produce products required for the Krebs cycle, including α-KG. Mutations in *IDH1* and *IDH2* lead to the conversion of α-KG to an oncometabolite, 2-hydroxyglutarate, inhibiting α-KG-dependent enzymatic reactions such as *TET2* DNA hydroxylation. Mutations in *IDH1* and *IDH2* occur in <10% of patients with MDS but are associated with an increased risk of transformation into AML. Both *IDH1* and *IDH2* have been shown to be associated with unfavorable prognosis, although the prognosis of *IDH2* remains controversial. *IDH1*, *IDH2,* and *TET2* are mutually exclusive but display an overlapping DNA hypermethylation signature [51]. Unlike *TET2*, *IDH1* and *IDH2* mutations are rare in CHIP. *TET2* mutations have been shown to have a positive correlation with *SRSF2* and *ZRSR2* [17]. Targeted therapy using IDH1 and IDH2 inhibitors, including ivosidenib and olutasidenib for IDH1 and enasidenib for IDH2, have been approved for the treatment of AML and are being investigated in MDS [52,53,54].

#### 3.2.3. Chromatin Modification

Polycomb genes encompass a family of protein complexes that have been discovered to impact chromatin structure and histone modification, resulting in the repression of gene transcription [55]. Polycomb proteins function within two multi-subunit protein complexes: polycomb repressive complex 1 (PRC1) and PRC2 with mono-ubiquinate histone H2A lysine 1199 and methylate (di- and tri-) histone H3 lysine 27, respectively. The core PRC2 complex comprises four components, EZH1/2, SUZ12, EED, and RBAP46/48, while the composition of PRC1 complexes exhibits more variability. Mutations involved in these complexes have frequently been identified in MDS, including *ASXL1*, *EZH2*, *KDM6A*, *SUZ12*, and *EED*. ASXL1 physically interacts with EZH2 and influences PRC2 recruitment in HSCs, and mutated ASXL1 results in the loss of interaction with the PCR2 complex [56]. KDM6A is an enzyme that facilitates the demethylation of H3K27. Thus, loss-of-function mutations of PRC2 components are thought to deregulate the normal program of hematopoiesis through repression of transcription key genes in hematopoietic stem/progenitor cells, contributing to positive selection.

*ASXL1* is the most frequently mutated gene in this category and is mutated in 15–20% of MDS cases [38,57]. Additionally, *ASXL1* is frequently mutated in CHIP as well as in 5–10% of AML cases. Mutations in *ASXL1* are more likely to coexist with other mutations, except with *SF3B1*, *DNMT3A,* and *IRF1* mutations for which they have negative correlations. *ASXL1* negatively impacts OS and increases the risk of relapse. The second most mutated gene that affects chromatin modification is *EZH2*, which is mutated in 5–10% of MDS patients. *TET2*, *RUNX1,* and *ASXL1* are most frequently mutated together with *EZH2* [38,58]. *EZH2* is located at 7q36.1, and in both AML and MDS, this region is frequently affected by loss of chromosome 7 or deletion 7q, which is associated with adverse outcomes. Indeed, studies have shown that, like *ASXL1*, *EZH2* is an independent unfavorable prognostic factor with progression to AML [15,59].

#### 3.2.4. Transcription Factors

Another group of mutations in MDS is the group of transcription factors. Transcription factors bind to specific DNA sequences, and mutations have been reported to cause impairment of differentiation and maintenance of the HSC. Somatic mutations are present in 10–15%, and common genes are *RUNX1*, *BCOR*, *ETV6*, *GATA2*, and *CIX1*. In addition, they can be present as germline mutations responsible for familial MDS/AML [60].

The RUNX1 transcription factor is a critical regulator of hematopoiesis [61]. Mutation of *RUNX1* disrupts the core-binding factor complex, leading to alteration of gene transcription. *RUNX1* accounts for about 10% of MDS cases (the third most frequently mutated gene in MDS) and is typically a subclonal mutation associated with unfavorable clinical outcomes and advanced disease. It is a common abnormality in therapy-related MDS [62]. Mutated *RUNX1* is frequently accompanied by additional mutations of the genes *ASXL1*, *SRSF2*, *TET2*, *SF3B1*, and *EZH2* and often co-exists with del(7)/del(7q) [61].

BCOR is a transcription factor that is a component of the PRC and encodes for a corepressor of BCL6. *BCOR* mutation is present in 5% of MDS patients and commonly co-occurs with *RUNX1* and *DNMT3A* mutations. Although the type of mutation may be important, *BCOR* mutations are associated with unfavorable outcomes [63,64]. *ETV6* is only present in less than 5% of the patients, and more than 30 fusion partner genes have been identified in a broad spectrum of hematologic malignancies, mainly T-ALL. In the IPSS-M, *ETV6* is significantly associated with worse OS and progression to AML after adjustment for IPSS risk groups (*p* = 0.04) [10,15]. GATA2 belongs to the GATA family of zinc finger transcription factors that are important for hematopoietic stem cell maintenance and differentiation. It is frequently associated with familial MDS but also occurs as a somatic mutation [65].

#### 3.2.5. Cohesin Complex

The cohesin ring is a conserved multimeric protein complex that is involved in sister chromatic cohesion during cell division, DNA repair, and transcription regulation. It is composed of two structural maintenance heterodimers, SMC1A and SMC3, that form a close loop with RAD21 and STAG1/STAG2 proteins. Moreover, they bind other regulatory molecules including NIPBL, PDS5b, and CTCF [66,67]. Mutations in the cohesin proteins lead to loss of function and have been identified in 10% of MDS patients as well as in other hematologic myeloid malignancies. Mutations lead to the loss of cohesin binding sites on chromatin that allow access to transcription factors. Particularly, *STAG2*, which is present in about 5% of the patients, has been associated with predicted poor survival [68].

#### 3.2.6. Signal Transduction

Mutations involved in signal transduction are less commonly associated with MDS compared to AML. Overall, they occur in 5–10% of the patients, with each individual mutation present in <5% of the cases. Activating mutations of tyrosine kinase and/or serine/threonine kinase results in constitutive activation of the JAK-STAT or RAS-MAPK pathway. Examples include *JAK2*, *CBL*, *NRAS*, and *NF1* of which *CBL*, a tumor suppressor with E3 ubiquitin ligase activity, is associated with reduced OS [69,70]. For *PTPN11*, *JAK2*, and *NF1*, no survival impact has been observed so far.

#### 3.2.7. TP53

TP53 is a tumor suppressor and a transcription factor and is the most frequently mutated gene in cancer [71]. It is mutated in approximately 10% of de novo MDS cases and in 25% of therapy-related MDS cases as well as in 20% of MDS patients with the 5q-deletion [72]. Moreover, about 50% of *TP53*-mutated patients have a complex karyotype. *TP53* mutation is associated with high-risk MDS, rapid transformation to AML, early relapse, and poor OS. *TP53* mutations define a separate disease entity per the 2022 ICC and also define a unique subgroup within patients with a complex karyotype [9,73]. Although most studies look at the impact of the presence/absence of mutated *TP53*, few studies have investigated the allelic status. Recently, Bernard et al. compared single-hit mutated *TP53* to multi-hit mutated *TP53* in a large cohort of >3300 MDS patients [74]. Their study found that multi-hit (i.e., meaning that cells have lost both copies of *TP53*) was more frequently found in patients with a complex karyotype, had fewer co-occurring mutations, and was associated with shorter OS and transformation into AML. Moreover, they also noted that single-hit *TP53* MDS patients were indistinguishable from nonmutant *TP53* MDS in terms of outcome and response to therapy.

## 4. Conclusions

MDS is a group of heterogeneous diseases arising from hematopoietic stem cells, which are characterized by ineffective hematopoiesis. The updated IPSS-M risk classification model has been improved from former models by incorporating molecular genetic mutation data for the first time, in combination with established recurrent cytogenetic aberrations, resulting in enhanced prognostic accuracy across all long-term clinical outcomes in MDS. Defining a more precise classification is crucial for the diagnostic approach, prognostication, and advanced therapeutic decision making.

## 5. Future Directions

The current treatment of MDS is based on the IPSS-R risk score stratifying patients into low-risk (i.e., IPSS-R very-low, low, or intermediate risk (IPSS-R score ≤ 3.5)) vs. high-risk MDS (i.e., R-IPSS intermediate, high, or very-high risk (R-IPSS score ≥ 4.0)) [14]. Treatment in low-risk patients is mostly focused on increasing blood counts and increasing quality of life, whereas high-risk patients are treated with either hypomethylating agents to prevent progression into AML or intensive chemotherapy and, depending on their response, followed by allogeneic stem cell transplantation. However, an increased understanding of the underlying molecular landscape of MDS has slowly led to the emergence of new agents that target molecular events or their downstream consequences, with several agents being in various stages of clinical development, either as a single agent or in combination with chemotherapy, hypomethylating agents, or molecular inhibitors. Ongoing efforts will likely result in the approval of molecular targeted drugs for specific subsets of patients.

Unraveling molecular driver events is furthermore pivotal to identify patients who are at high risk for progression into AML, as these should be aggressively treated and referred for transplant early in the course of their disease. It also aids in the identification of patients who likely will do poorly with current treatments and for whom enrollment in clinical trials should be considered as well as in informing the development of new treatment strategies. Conversely, accurate risk stratification models can lead to the recognition of low-risk patients who can be treated with supportive therapies, sparing them from unnecessary exposure to toxic treatment. Future studies are warranted to investigate the “significance” of CCUS and should answer the question of when we should intervene for patients with CCUS to prevent progression to MDS or AML.

**Table 3 cells-12-00627-t003:** Most common driver genes in patients with MDS.

	Frequency (%)	Location	Prognostic Impact	Function	Ref
RNA splicing (40–50%)					
*SF3B1*	25–30% Frequently associated with MDS-RS	2q33	Favorable Unfavorable if combined with del 5q.	Subunit 1, RNA-splicing factor 3b complex, part of U2 small nuclear ribonucleoprotein complex (snRNP)	[10,40,41]
*SRSF2*	15% Higher frequently in chronic myelomonocytic leukemia (50%)	17q25	Unfavorable OS, high-risk transformation into AML	Serine/arginine (SR) rich splicing factor 2, family of pre-mRNA splicing factors	[10,44,75]
*U2AF1*	10–15%	21q22	Unfavorable OS, high-risk transformation into AML	Heteromeric with U2AF2 to form U2 auxiliary factor (U2AF), recruits U2 snRNP. Pre-mRNA splicing factor.	[10,46]
*ZRSR2*	5–10%	Xp22	Unclear	Zinc finger RNA-binding associated with U2. 3′ intron splice site recognition.	[44,76]
*U2AF2*	Rare	19q13	Unfavorable, associated high-risk MDS and AML	Heteromeric with U2AF1 that forms U2AF	[77,78]
DNA methylation (30–40%)					
*TET2*	20–30%	4q24	Unclear	Alpha ketoglutarate-dependent dioxygenase	[10,15,17]
*DNMT3A*	15%	2q23	Unfavorable, associated risk transformation into AML	DNA methyltransferase 3A, catalyzes transfer methyl groups to cytosine residue in CpG dinucleotides	[10,17]
*IDH2*	5%	2q33	Unclear, studies suggest unfavorable prognosis	NADPH-dependent isocitrate dehydrogenase	[10,69]
*IDH1*	2%	15q26	Unfavorable	NADPH-dependent isocitrate dehydrogenase	[79,80]
Chromatin modification (20%)					
*ASXL1*	15–20%	20q11	Unfavorable	Polycomb group protein, chromatin-binding protein	[10,38,56]
*EZH2*	5–10%	7q36	Unfavorable	Polycomb group protein, histone methyl transferase	[15,38,59]
*KDM6A*	<5%	Xp11	Unclear	Polycomb group protein, lysine demethylation	[10]
*EED*	<5%	11q14	Unclear	Polycomb group protein, histone methyl transferase	[10]
Transcription factors (10–15%)					
*RUNX1*	10%	21q22	Unfavorable	Transcription factor, core-binding factor complex	[10,15,16]
*BCOR*	5%	Xp11	Unfavorable	Transcription factor, polycomb complex protein	[10,15,16,63]
*ETV6*	<5%	12p13	Unfavorable	ETS family transcription factor	[10,15]
*GATA2*	<5%	3q21	Unfavorable	Zinc finger transcription factor	[10,62]
Cohesin (10%)					
*STAG2*	5%	Xq25	Unfavorable	Component cohesin complex	[10,68]
*RAD21*	<5%	8q24	Unclear	Component cohesin complex	[68]
Signal transduction (5–10%)					
*JAK2*	<5%	9p24	Unclear	Tyrosine kinase, JAK-STAT pathway	[15]
*CBL*	5%	11q23	Unfavorable	Tyrosine kinase, E3 ubiquitin-protein ligase	[15,70]
*NRAS*	<5%	1q13	Unfavorable	Tyrosine kinase, RAS-MAPK pathway	[10,81]
*KRAS*	<5%	12p12	Unfavorable	Tyrosine kinase, RAS-MAPK pathway	[10,81]
*FLT3-ITD*	<5%	13q12	Unfavorable	Class III family receptor tyrosine kinase	[10]
*KIT*	<5%	4q11-12	Unclear	Class III family receptor tyrosine kinase	[10]
*PTPN11*	<5%	12q24	Unclear	Protein phosphatase	[16]
Tumor suppressor (5–10%)					
*TP53*	10%, 50% in complex karyotype	17p13	Unfavorable	Tumor suppressor, transcription factor	[10,74]
*WT1*	5%	11p13	Unfavorable, associated with disease progression to AML	Tumor suppressor, transcription factor	[10,82]
*PHF6*	5%	Xq26-27	Unfavorable	Tumor suppressor, epigenetic transcriptional regulator	[83]

## Figures and Tables

**Figure 1 cells-12-00627-f001:**
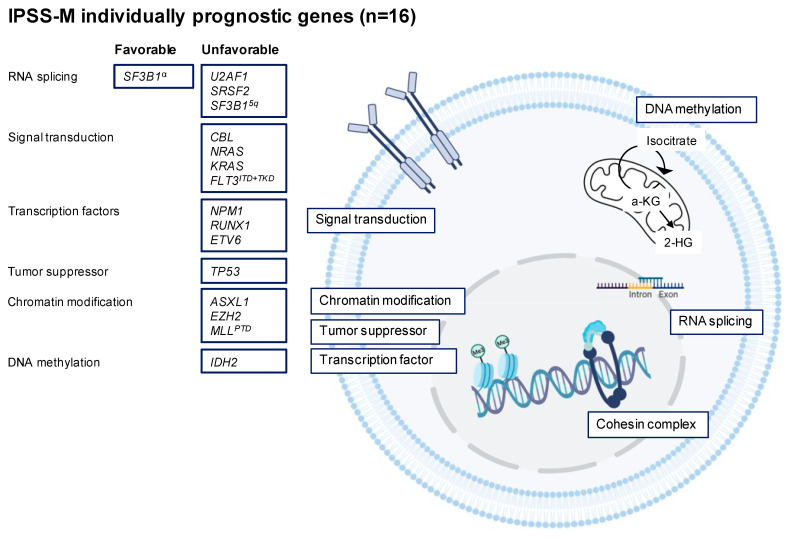
Representation of the individually prognostic mutations incorporated in the IPSS-M prognostication model.

**Table 1 cells-12-00627-t001:** MDS per the 2022 WHO and ICC classifications.

Morphologically Defined	Genetically Defined	WHO 2022	ICC 2022
Ring sideroblasts		MDS with low blasts and *SF3B1* mutation. Detection of ≥15% ring sideroblasts may substitute for *SF3B1* mutation.	None
Number of dysplastic lineages		Number of dysplastic lineages is no longer included in WHO 2022.	MDS, NOS without dysplasia (−7/del(7q) or complex, and any mutations except multihit *TP53* or *SF3B1* ≥ 10% VAF)
			MDS, NOS with single lineage dysplasia (any cytogenetics, except for MDS-de(5q), and any mutations except multihit *TP53* and not meeting criteria MDS *SF3B1*)
			MDS, NOS with multilineage dysplasia (any cytogenetics, except for MDS-de(5q), and any mutations except multi-hit *TP53* and not meeting criteria MDS *SF3B1*)
Blasts%		MDS with low blasts (MDS-LB): <5% bone marrow (BM) and <2% peripheral blood (PB).	None
		MDS, hypoplastic (MDS-h): <25% BM cellularity, age-adjusted	None
		MDS with increased blasts (MDS-IB): MDS-IB1: 5–9% BM or 2–9% PB MDS-IB2: 10–19% BM or 5–19% PB or Auer rods	MDS with excess blasts (MDS-EB): 5–9% BM or 2–9% PB, any cytogenetics or mutations, except multihit *TP53*.
		MDS with fibrosis (MDS-f): 5–19% BM or 2–19% PB	MDS/AML: 10–19% BM or PB blasts with any cytogenetics, except for AML-defining, and any mutations except for *NPM1*, bZIP *CEBPA,* and *TP53*
Genetically defined Subtypes	Isolated 5q	MDS with low blasts and isolated 5q deletion (MDS-5q): 5q deletion alone or with 1 other abnormality other than monosomy 7 or 7q deletion	MDS with del(5q): del(5q) with up to one additional, except for −7/del(7q), with any mutations except multihit *TP53*
	*SF3B1*	MDS with low blasts and *SF3B1*: mutation in the absence of 5q deletion, monosomy 7, or complex karyotype	MDS with mutated SF3B1: *SF3B1* (≥10% VAF) without multihit *TP53* or *RUNX1*, with any cytogenetics except for isolated del(5q), −7/del(7q), abn3q26.2, or complex.
	*TP53*	MDS with biallelic *TP53* inactivation (MDS-biTP53): two or more *TP53* mutations or 1 mutation with evidence of *TP53* copy number loss or copy neutral loss of heterozygosity. In the presence of ≤20% BM or PB blasts	Myeloid neoplasm with mutated *TP53* (MDS-*TP53*, *MDS/AML-TP53*) Defined as 2 distinct *TP53* mutations (each VAF > 10%) OR a single *TP53* mutation with (1) 17p deletion on cytogenetics; (2) VAF of >50%; or (3) copy-neutral LOH at the 17p *TP53* locus.

**Table 2 cells-12-00627-t002:** IPSS and IPSS-R classification. Table adapted from Greenberg et al. published in 1997 and Greenberg et al. 2012 [11,12].

	Score						
Variable	0	0.5	1.0	1.5	2.0	3.0	4.0
IPSS							
Bone marrow blast (%)	<5%	5–10%		11–20%	21–30%		
Karyotype ^†^	Good	Intermediate	Poor				
Cytopenias ^††^	0/1	2/3					
IPSS-R							
Cytogenetics ^†††^	Very good		Good		Intermediate	Poor	Very poor
Bone marrow blast (%)	≤2%		>2 to <5%		5–10%	>10%	
Hemoglobin (g/dL)	≥10		8 to <10	<8			
Platelets (cells/µL)	≥100	50–100	<50				
Absolute neutrophil count (cell/µL)	≥0.8	<0.8					

^†^ Karyotype. Good: normal, -Y, del (5q), and del (20q); poor: complex (≥3 abnormalities) and abnormal chromosome 7; intermediate: all others. ^††^ Cytopenia definitions. Red blood cells: Hemoglobin <10 g/dL (100 g/L); white blood cells: absolute neutrophil count <1800/µL; platelets: platelet count <100,000/µL. ^†††^ Cytogenetic definitions. Very good: –Y and del(11q); good: normal, del(5q), del(12p), del(20q), and double including del(5q); intermediate: del(7q), +8, +19, i(17q), any other single, double not including del(5q) or –7/del(7q), or independent clones; poor: –7, inv(3)/t(3q)/del(3q), double including –7/del(7q), and complex (3 abnormalities); very poor: complex (>3 abnormalities).

## Data Availability

Not applicable.

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
