# Peer review of "Molecular Drivers of Myelodysplastic Neoplasms (MDS)—Classification and Prognostic Relevance"

_cells, 2023, doi:10.3390/cells12040627_

Round 1

Reviewer 1 Report

This paper deals with karyotipic and molecular alterations in MDS, an important issue considering the new IPSS-M classification recently proposed. Manuscript is clear and readible, even if the tables are quite crowded and perhaps deserve some graphical editing (for example, Table 2 could be splitted).

My tentative comments to improve the already high quality of the paper are as follows:

1) I would add a Table with IPSS-R criteria

2) Monosomal karyopype is an important feature to report in the paragraph of cytogenetic abnormalities

3) According to literature, median age of 5q-syndrome is higher than 65-70 years: I suggest at least to change in ">70 years"

Minor comments regard grammar of the test, in which some check spells are required: 

line 25: "dependents" is probably "depend"

lines 80 - 86: the phrase should be rewritten (The disease was called refractory anemia by the FAB classification, WHICH categorized MDS on the basis of morphologic criteria and percentage of myeloid blasts into....."

lines 104 and 108 could be probably evidencied in bold

In the figure 1 "Signal transcription" should be changed in "Signal transduction"

line 189: "of" should be erased

line 220: "is" should be included between "therefore" and "associated"

line 257: "mutated" should be changed into "frequent" or "common"

line 402: "for" or "from"?

Author Response

Thank you very much for your valuable time. 

Reviewer 2 Report

The authors present an updated review on several current topics in MDS. The review is easy to read and adequately addresses the issues discussed.

I have several comments I would like to share with the authors.  They are listed in order of appearance in the text.

1) "2.1 French-American-British (FAB)" paragraph: line 82. Delete the "and" in front of "into".

2) "2.2 WHO Classification" paragraph: line 98. "as CHIP or CCUS, defined as CHIP" should be "as CHIP or CCUS, defined as CCUS".

The authors comment that "the threshold of dysplasia was changed to 10%". The 10% threshold for considering a line dysplastic has not changed in WHO 2022 with respect to previous WHOs.

The list of new MDS categories according to WHO 2022 could be improved with a table.

2) "2.3 International Consensus Classification (ICC)" section. Line 126. In the ICC classification they use multihit TP53 instead of bi-allelic TP53.

Before the comparative table between the two classifications, I would appreciate a table with the ICC 2022 classification.

3) "International Prognostic Scoring System (IPSS)" section. I would appreciate a comment from the authors on the difficult prognostic consideration of patients classified in the IPSS-R intermediate risk category. It is not clear from the conception of the index whether these patients should be considered low or high risk. I would appreciate it if they could provide literature on the subject and comment on the IWG's decision to consider patients with >3.5 IPSS-R points as high risk patients.

Line 144. "Showing that genetic mutations harbors prognostic information". Please introduce several references in this regard. I would appreciate a comment on the discordance between the number of genes with prognostic value detected in previous studies (Nazha A, Bejar R, ...) and the large number of genes detected in the IPSS-M. Why do you think that such underrepresented genes in MDS as FLT3, NPM1 or MLL-PTD maintain an independent prognostic value? Could these discordances be explained by the different statistical methodology used in the IPSS-M? Please issue a reasoned comment on all of the above.

Figure 1. All genes in italics. MLL should be MLL-PTD.

4) "3.2.1. RNA splicing". Line 253. ZRSR2 instead of ZRZR1. Line 283. "splice" instead of "slice".

A comment on the subtypes of DNMT3A mutations in CHIP (few in the DNA methyltransferase domain) and myeloid neoplasms (many in the DNA methyltransferase domain, especially R882) would be interesting.

Thank you very much for the effort. It has been a pleasure to review your article.

Author Response

(The authors gave the same response as above.)
